# Risk-Averse Zero-Order Trajectory Optimization

**Marin Vlastelica**[1*]**, Sebastian Blaes**[1*]**, Cristina Pinneri**[1,2] **and Georg Martius**[1]
[1]Autonomous Learning Group, Max Planck Institute for Intelligent Systems, Tübingen, Germany
[2]Department of Computer Science, ETH Zurich and Max Planck ETH Center for Learning Systems
{mvlastelica,sblaes,cpinneri,gmartius}@tuebingen.mpg.de

**Abstract:** We introduce a simple but effective method for managing risk in zero-order trajectory optimization that involves probabilistic safety constraints and balancing of optimism in the face of epistemic uncertainty and pessimism in the face of aleatoric uncertainty of an ensemble of stochastic neural networks. We empirically validate that the separation of uncertainties is essential to performing well with data-driven MPC approaches in uncertain and safety-critical control environments.

**Keywords:** zero-order trajectory optimization, CEM, MPC, aleatoric, epistemic, safety, uncertainty, model learning

## 1    Introduction

Data-driven approaches to sequential decision-making are becoming increasingly popular [1, 2, 3, 4]. They hold the promise of reducing the number of prior assumptions about the system that are imposed by traditional approaches that are based on nominal models.

Such approaches come in several different flavors [5]. Model-free approaches attempt to extract closed-loop control policies directly from data, while model-based approaches rely on a learned model of the dynamics to either generate novel data to extract a policy or to be used in a model-predictive control fashion (MPC). This work belongs to the latter line of work.

Model-based methods have several advantages over pure model-free approaches. Firstly, humans tend to have a better intuition on how to incorporate prior knowledge into a model rather than into a policy or value function. Secondly, most model-free policies are bounded to a specific task, while models are task-agnostic and can be applied for optimizing arbitrary cost functions, given sufficient exploration.

Nevertheless, learning models for control comes with certain caveats. Traditional MPC methods require the model and cost function to permit a closed-form solution which restricts the function class prohibitively. Alternatively, gradient-based iterative optimization can be employed, which allows for a larger class of functions but typically fails to yield satisfactory solutions for complicated function approximators such as deep neural network models. In addition, calculating first-order or even second-order information for trajectory optimization tends to be computationally costly, which makes it hard to meet the time constraints of real-world settings. This motivates the usage of zero-order methods, i.e gradient-free or sample-based, such as the Cross-entropy Method (CEM) that do not rely on gradient information but are efficiently parallelizable.

Many methods relying on a learned model and zero-order trajectory optimizers have been proposed [6, 7, 8], but all share the same problem: compounding of errors through auto-regressive model prediction. This naturally brings us to the question of how can we effectively manage model errors and uncertainty to be more data-efficient and safe. Arguably, this is one of the main obstacles to applying data-driven model-based methods to the real world, e.g. to robotics settings.

In this work, we introduce a risk-averse zero-order trajectory optimization method (RAZER) for managing errors and uncertainty in zero-order MPC and test it on challenging scenarios (Fig. 1). We argue that it is essential to differentiate between the two types of uncertainty in the model-predictive setting: the aleatoric uncertainty arising from inherent noise in the system and epistemic uncertainty arising from the lack of knowledge [9, 10]. We measure these uncertainties by making use of probabilistic ensembles with trajectory sampling similar to PETS [6]. Our contributions can be

---

*equal contribution

5th Conference on Robot Learning (CoRL 2021), London, UK.

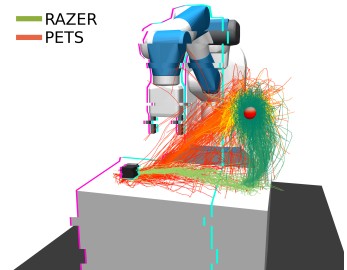 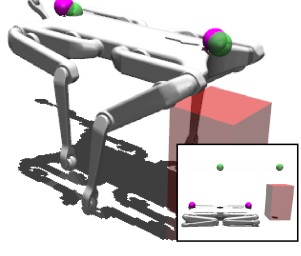 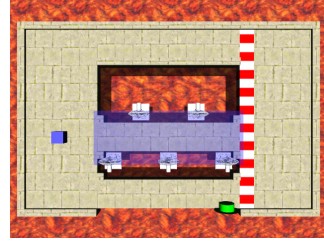

(a) Noisy-FetchPickAndPlace  (b) Solo8-LeanOverObject  (c) BridgeMaze

Figure 1: Environments considered for uncertainty-aware planning. Code and videos are available at https://martius-lab.github.io/RAZER/

summarized as follows: (i) method for separation of uncertainties in probabilistic ensembles (termed PETSUS); (ii) efficient use of aleatoric and epistemic uncertainty in model-based zero-order trajectory optimizers; (iii) a simple but practical approach to probabilistic safety constraints in zero-order MPC.

## 2   Related Work

**Uncertainty Estimation.**  In the typical model-based reinforcement learning (MBRL) setting, the true transition dynamics function is modeled through an approximator. Impressive results have been achieved by both parametric models [11, 12, 13, 14], such as neural networks, and nonparametric models [15, 16, 17, 18], such as Gaussian Processes (GP). The latter inspired seminal work on the incorporation of the dynamics model's uncertainty for long-term planning [18, 19]. However, their usability is limited to low-data, low-dimensional regimes with smooth dynamics [20, 21], which is not ideal for robotics applications. Alternative parametric approaches include ensembling of deep neural networks, used both in the MBRL community [6, 22], and outside [23, 24]. In particular, ensembles of *probabilistic* neural networks established state-of-the-art results [6], but focus mainly on estimating the expected cost and disregard the underlying uncertainties. In comparison, we propose a treatment of the resulting uncertainties of the ensemble model.

**Zero-order MPC.**  The learned model can be used for policy search like in PILCO [25, 18, 19, 26] or for online model-predictive control (MPC) [27, 28, 6]. In this work, we do planning in an MPC fashion and employ a zero-order method as a trajectory optimizer, since they have shown to be less likely to get stuck in local minima and make an explicit treatment of the uncertainty in the cost possible. Specifically, we consider a sample-efficient implementation of the Cross-Entropy method [29, 30] introduced in [31].

**Safe MPC.**  Separating the sources of uncertainty is of particular importance for applications directly affecting humans' safety, as self-driving cars, elderly care systems, or in general any application that involves a physical interaction between agents and humans. Disentangling epistemic from aleatoric uncertainty allows for separate optimization of the two, as they represent semantically different objectives as per definition. Extensive research on uncertainty decomposition has been done in the Bayesian setting and the context of safe policy search [32, 33, 34, 35], MPC planning [36, 37, 38], and distributional RL [39, 40]. On the other hand, a state-of-the-art baseline for ensemble learning like PETS [6], despite estimating uncertainty, only optimizes for the *expected* cost during action evaluation. Our work aims at filling this gap by explicitly integrating the propagated uncertainty information in the zero-order MPC planner.

## 3   Method

Our approach concerns itself with the efficient usage of uncertainties in zero-order trajectory optimization and is therefore generally applicable to such optimizers. We are interested in modeling noisy system dynamics $x_{t+1} = f(x_t, u_t, w(x_t, u_t))$ where $f$ is a nonlinear function, $x_t$ the observation vector, $u_t$ applied control input and $w(x_t, u_t)$ a noise term sampled from an arbitrary distribution.

Consequently, in the absence of prior knowledge about the function $f$, the system needs to be modeled by a complex function approximator such as a neural network. Furthermore, we are interested in managing uncertainties based on our fitted model, which is erroneous. To this end, we

use stochastic ensembles of size $K$, where the output of each model $\boldsymbol{\vartheta}^k(x_t, u_t)$ are parameters of a normal distribution depending on input observation $x_t$ and control $u_t$. As a by-product, our auto-regressive model prediction based on sequence of control inputs $\boldsymbol{u}$ becomes a predictive distribution over trajectories $\tau = (x_0, u_0, x_1, u_1 \ldots)$; $\psi^\tau(x_t, \boldsymbol{u}) := p(\tau | x_t, \boldsymbol{u}; \theta)$ where $\theta$ denotes the parameters of the ensemble. For convenience, from this point onward we will differentiate between multiple usages of $\psi^\tau$. We denote with $\psi^x_{\Delta t}$ the distribution $p(x_{t+\Delta t} | x_t, \boldsymbol{u}_{t:t+\Delta t-1}; \theta)$ over states at time step $t + \Delta t$, $\psi^{\boldsymbol{\vartheta}}_{\Delta t}$ the distribution over the Gaussian parameter outputs $p(\boldsymbol{\vartheta}_{t+\Delta t} | x_t, \boldsymbol{u}_{t:t+\Delta t-1}; \theta)$ at time step $t + \Delta t$ of the planner.

## 3.1 Planning and Control

To validate our hypothesis that accounting for uncertainty in the environment and model prediction is essential to develop risk-averse policies, we use the Cross-Entropy Method (CEM) with improvements suggested in Pinneri et al. [31]. Accordingly, at each time step $t$ we sample a finite number of control sequences $\boldsymbol{u}$ for a finite horizon $H$ from an isotropic Gaussian prior distribution which we evaluate from the state $x_t$ using an auto-regressive forward-model and the cost function. The sampling distribution is refitted in multiple rounds based on the best performing trajectories. After this optimization step, the first action of the mean of the fitted Gaussian distribution is executed. Since this approach utilizes a predictive model for a finite horizon at each time step, it naturally falls into the category of MPC methods.

Although we use CEM, our approach of managing uncertainty can generically be applied to other zero-order trajectory optimizers such as MPPI [28] by a modification of the trajectory cost function.

## 3.2 The Problem of Uncertainty Estimation

Since we have a stochastic model of the dynamics, at the model prediction time step $t$ we observe a distribution over potential outcomes. Indeed, since our model outputs are parameters of a Gaussian distribution, with auto-regressive predictions we end up with a distribution over possible Gaussians for a certain time step $t$.

Given a sampled action sequence $\boldsymbol{u}$ and the initial state $x_t$ we observe a distribution over trajectories $\psi_\tau$. To efficiently sample from the trajectory distribution $\psi_\tau$ we use the technique introduced by Chua et al. [6] (PETS) which involves prediction particles that are sampled from the probabilistic models and randomly mixed between ensemble members at each prediction step. In this way, the sampled trajectories are used to perform a Monte Carlo estimate of the expected trajectory cost $\mathbb{E}_{\tau \sim \psi^\tau}[c(\tau)]$. However, this does not take the properties of $\psi^\tau$ into account, which might be a high-entropy distribution and may lead to very risky and unsafe behavior. In this work, we alleviate this by looking at the properties of $\psi^\tau$, i.e. different kinds of uncertainties arising from the predictive distribution.

## 3.3 Learned Dynamics Model

We learn a dynamics model $f_\theta$ that approximates the true system dynamics $x_{t+1} = f(x_t, u_t, w(x_t, u_t))$. As a model class, we use an ensemble of neural networks with stochastic outputs as in Chua et al. [6]. Each model $k$, parameterizes a multivariate Gaussian distribution with diagonal covariance, $f^k_\theta(x_t, u_t) = \mathcal{N}(x_{t+1}; x_t + \mu^k_\theta(x_t, u_t), \Sigma^k_\theta(x_t, u_t))$ where $\mu^k_\theta(\cdot, \cdot)$ and $\Sigma^k_\theta(\cdot, \cdot)$ are model functions outputting the respective parameters.

Iteratively, while interacting with the environment, we collect a dataset of transitions $\mathcal{D}$ and train each model $k$ in the ensemble by the following negative log-likelihood loss on the Gaussian outputs:

$$\mathcal{L}(\theta, k) = \mathbb{E}_{x_t, u_t, x_{t+1} \sim \mathcal{D}} \Big[ - \log \mathcal{N}(x_{t+1}; x_t + \mu^k_\theta(x_t, u_t), \Sigma^k_\theta(x_t, u_t)) \Big] \tag{1}$$

In addition, we use several regularization terms to make the model training more stable. We provide more details on this in Suppl. A.

## 3.4 Separation of Uncertainties

In the realm of parametric estimators, two uncertainties are of particular interest. *Aleatoric* uncertainty is the kind that is irreducible and results from inherent noise of the system, e.g. sensor noises in robots. On the other hand, we have *epistemic* uncertainty resulting from lack of data or knowledge which is

reducible. This begs the question, how can we separate these uncertainties given an auto-regressive dynamics model $\bar{f}_\theta$? The way that we efficiently sample from $\psi^\tau$ is by mixing sampled prediction particles. This process is illustrated by the red lines in Fig. 2.

Simple model prediction disagreement is not a good measure for *aleatoric* uncertainty since it can be entangled with epistemic uncertainty. Given our how we model the system dynamics, we measure *aleatoric* uncertainty as entropy of the predicted normal distributions across ensemble members. More concretely, given a sampled particle state $\tilde{x}_t$, we define the estimated aleatoric uncertainty for an ensemble member associated to particle $b$ at time step $t$ as:

$$\mathfrak{A}_b(x|\tilde{x}_t, u_t) = \mathcal{H}_{x \sim \psi^x_{\Delta t, b}}(x) \tag{2}$$

Where $\psi^x_{\Delta t, b}$ is the output distribution of ensemble model based on inputs $\tilde{x}_t$, $u_t$. Since in the end we are interested in the aleatoric uncertainty incurred from applying the action sequence $\boldsymbol{u}$ from initial state $x_t$, the quantity of interest for us is the expected aleatoric uncertainty for time slice $t$:

$$\mathfrak{A}(x|u_t) = \mathbb{E}_{\tilde{x}_b \sim \psi^x_{\Delta t}}\left[\mathfrak{A}_b(x|\tilde{x}_b, u_t)\right] \tag{3}$$

Intuitively, because we only have access to the ensemble for sampling, we take a time-slice in the sampled trajectories from $\psi^\tau$ and compute the output entropies. Moreover, since we assume a Gaussian 1-step predictive distribution this is an expectation over differential Gaussian entropy. An alternative way of computation which we also explore in this work is calculating the expected particle variance for time slice $t + 1$ of the prediction horizon:

$$\text{Var}^{\mathfrak{A}}_{t+1} = \frac{1}{B}\sum_{b=1}^{B} \Sigma^k_\theta(\tilde{x}_{t,b}, u_t) \tag{4}$$

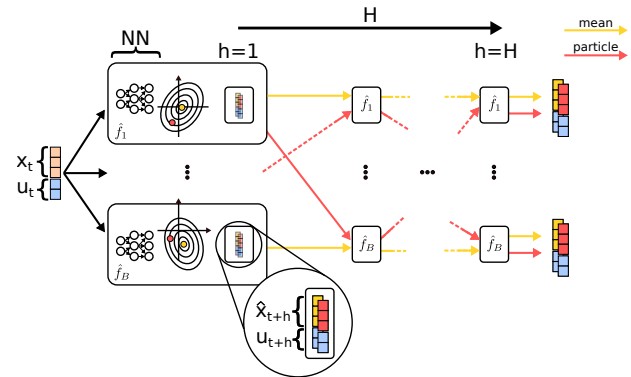

Note that $\Sigma^k_\theta(\tilde{x}_{t,b}, u_t)$ outputs the covariance of the prediction at $t + 1$. For estimating the *epistemic* uncertainty, one would be tempted to look at the disagreement between ensemble models in parameter space $\text{Var}[\theta]$, but this is not completely satisfying, since neural networks tend to be over-parametrized and variance within the ensemble still may exist albeit the optimum has been reached by all ensemble models. An alternative would be to calculate the Fisher information metric $\mathcal{I} := \text{Var}[\nabla_\theta \log \mathcal{L}(x_{t+1}|x_t, u_t)]$ where $\mathcal{L}$ denotes the likelihood function, but this tends to be expensive to compute.

Figure 2: Probabilistic Ensembles with Trajectory Sampling and Uncertainty Separation (PETSUS)

Given the assumption of local Gaussianity, the true epistemic uncertainty for this case is the predictive entropy over the Gaussian parameters $\boldsymbol{\vartheta}$ at time step $t + h$.

$$\mathfrak{E}(x_t, \boldsymbol{u}_{t:t+\Delta t-1}) = \mathcal{H}_{\psi^\vartheta_{\Delta t}}(\boldsymbol{\vartheta} \mid x_t, \boldsymbol{u}_{t:t+\Delta t-1}) \tag{5}$$

It is easy to verify that this quantity is 0 given perfect predictions of the model. Note that, because of auto-regressive predictions of a nonlinear model, this is a very difficult object to handle. Nevertheless, since our predictive distribution $p(x|x_t, u_t; \boldsymbol{\vartheta})$ is parametrized by model output, we may utilize disagreement in $\boldsymbol{\vartheta}_t$ to approximate $\mathfrak{E}$. To get correct estimations, we need to propagate mean predictions $\bar{x}$ in addition to the particles as illustrated as the yellow lines in Fig. 2. We quantify epistemic uncertainty as ensemble disagreement at time step $t$:

$$\text{Var}^{\mathfrak{E}}(x_{t+1}) = \text{Var}^e[\mu^k_\theta(\bar{x}_t, u_t)] + \text{Var}^e[\Sigma^k_\theta(\bar{x}_t, u_t)] \tag{6}$$

where $\text{Var}^e$ is the empirical variance over the $k = 1 \ldots K$ ensembles.

### 3.5 Probabilistic Safety Constraints

When applying data-driven control algorithms to real systems, safety is of utmost importance. In the realm of zero-order optimization, safety constraints can be easily introduced by putting an infinite

cost on constraint-violating trajectories. Nevertheless, we are dealing with erroneous stochastic nonlinear models which lead to nontrivial predictive distributions of future states, based on the control sequence $\boldsymbol{u}$. For this reason, we want to control the risk of violating the safety constraints that we, as practitioners, are willing to tolerate. If we denote the observation space as $\mathbb{X}$, given a violation set $\mathbb{C} \subset \mathbb{X}$, we define the probability of the control sequence $\boldsymbol{u}$ to enter the violation set at time $t + \Delta t$ as $p(x \in \mathbb{C} \mid x_t, \boldsymbol{u}) = \int_{x \in \mathbb{C}} \psi_{\Delta t}^x(x \mid x_t, \boldsymbol{u})$. In practice, it is hard to compute this integral efficiently, since our distribution $\psi_{\Delta t}^x$ is nontrivial as a result of nonlinear propagation of uncertainty. Furthermore, the violation set $\mathbb{C}$ might not have the structure necessary to allow an efficient solution to the integral, in which case one needs to resort to Monte Carlo estimation.

To simplify computation and gain speed, we consider box violation sets resulting in each dimension of $x$ being constrained to be outside of $[a, b] \in \{a, b \mid a, b \in \mathbb{R}^2, a < b\}$. By performing moment matching by a Gaussian in each time-slice $\psi_{\Delta t}^x$, the probability of ending up in state $x$ at time step $t + \Delta t$ is given by integrating $\mathcal{N}(x; \mu_{t+\Delta t}, \Sigma_{t+\Delta t})$, where $\mu$ and $\Sigma$ are estimated by Monte Carlo sampling. If we further assume a diagonal covariance $\Sigma$, this integral can be deconstructed into $d$ univariate Gaussian integrals, which can be computed fast and in closed form. Hence, the probability of a constraint violation happening at time step $t$ is defined by:

$$p(x \in \mathbb{C} \mid x_t, \boldsymbol{u}) = \prod_{i=0}^{d} \int_{x \in \mathbb{C}} \mathcal{N}(x^i; \mu_{t+\Delta t}^i, \sigma_{t+\Delta t}^i) \tag{7}$$

### 3.6 Implementing Risk-Averse ZERo-Order Trajectory Optimization (RAZER)

We assume the task definition is provided by the cost $c(x_t, \boldsymbol{u})$. For trajectory optimization, we start from a state $x_t$ and predict with an action sequence $\boldsymbol{u}$ the future development of the trajectory $\tau$. Along this trajectory, we want to compute a single cost term which is conveniently defined as the expected cost of all particles $\tilde{x}$ summed over the planning horizon $H$:

$$c(x_t, \boldsymbol{u}) = \sum_{\Delta t=1}^{H} \frac{1}{B} \sum_{b=1}^{B} c(\tilde{x}_{t+\Delta t}^b, u_{t+\Delta t}). \tag{8}$$

The optimizer, in our case CEM, will optimize the action sequence $\boldsymbol{u}$ to minimize the cost in a probabilistic sense, i.e. $p(\boldsymbol{u} \mid x) \propto \exp(-\beta\, c(x, \boldsymbol{u}))$ where $\beta$ reflects the strength of the optimizer (the higher the more likely it finds the global optimum). To make the planner uncertainty-aware, we need to make sure it avoids unpredictable parts of the state space by making them less likely. Using the aleatoric uncertainty provided by PETSUS Eq. 4, we define the aleatoric penalty as

$$c_{\mathfrak{A}}(x_t, \boldsymbol{u}) = w_{\mathfrak{A}} \cdot \sum_{\Delta t=1}^{H} \sqrt{\mathrm{Var}_{t+\Delta t}^{\mathfrak{A}}}, \tag{9}$$

where $w_{\mathfrak{A}} > 0$ is a weighting constant. The larger the aleatoric uncertainty, the higher the cost.

To guide the exploration to states where the model has epistemic uncertainty Eq. 6 (due to lack of data), we use an epistemic bonus:

$$c_{\mathfrak{E}}(x_t, \boldsymbol{u}) = -w_{\mathfrak{E}} \cdot \sum_{\Delta t=1}^{H} \sqrt{\mathrm{Var}_{t+\Delta t}^{\mathfrak{E}}}, \tag{10}$$

where $w_{\mathfrak{E}} > 0$ is a weighting constant. To be able to operate on a real system, the most important part is to adhere to safety constraints. As formulated in Eq. 7, the predicted safety violations need to be uncertainty aware, independent of the source of uncertainty. We integrate this into the planning method by adding:

$$c_{\mathfrak{S}}(x_t, \boldsymbol{u}) = w_{\mathfrak{S}} \cdot \sum_{\Delta t=1}^{H} [\![ p(\hat{x}_{t+\Delta t} \in \mathbb{C}) > \delta ]\!] \tag{11}$$

where $[\![\cdot]\!]$ is Iverson bracket. and $w_{\mathfrak{S}}$ is either a large penalty $c_{\max}$ or 0 to disable safety. An alternative for implementing safety constraints into CEM is by changing the ranking function [41]. The overall algorithm used in a model-predictive control fashion is outlined in Suppl. B.

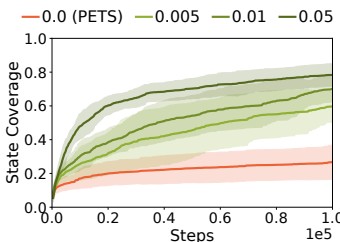
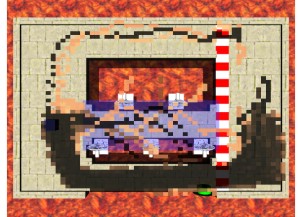
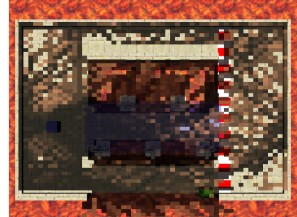

(a) State space exploration over time depending on epistemic bonus ($w_{\mathfrak{e}}$).

(b) State space coverage with $w_{\mathfrak{e}} = 0$.

(c) State space coverage with $w_{\mathfrak{e}} = 0.05$.

Figure 3: Active learning setting: The epistemic bonus allows RAZER to seek states for which no or only little training data exists (a,c). Means and standard deviations for (a) were computed over 5 runs. PETS overfits to a particular solution (b). In (b) and (c), the brightness of the dots is proportional to the time when they were first encountered.

## 4 Experiments

We study our uncertainty-aware planner in 4 continuous state and action space environments and compare to naively optimizing the particle-based estimate of the expected cost similarly to Chua et al. [6]. We start by giving a description of the environments.

**BridgeMaze** This toy environment (see Fig. 1c) was specifically designed to study the different aspects of uncertainty independently. The agent (blue cube) starts on the left platform and has to reach the goal platform on the right. To reach the goal platform, the agent has to move over one of three bridges without falling into the lava. The upper bridge is safeguarded by walls; hence, it is the safest path to the goal but also the longest. The lower bridge has no walls and therefore is more dangerous for an unskilled agent to cross but the path is shorter. The middle bridge is the shortest path to the goal. However, randomly appearing strong winds perpendicular to the bridge might cause the agent to fall off the bridge with some probability, making this bridge dangerous.

**Noisy-HalfCheetah** This environment is based on *HalfCheetah-v3* from the OpenAI Gym toolkit. We introduce aleatoric uncertainty to the system by adding Gaussian noise $\xi \sim \mathcal{N}(\mu, \sigma^2)$ to the actions when the forward velocity is above 6. The action noise translates into a non-Gaussian and potentially very complicated state space noise distribution that makes the control problem very challenging.

**Noisy-FetchPickAndPlace** Based on the *FetchPickAndPlace-v1* gym environment. Additive action noise is applied to the gripper so that its grip on the box might become tighter or looser. The noise is applied for $x$-positions $< 0.8$ which is illustrated in Fig. 1a by a blue line causing the agent to drop the box with high probability if it tries to lift the box too early.

**Solo8-LeanOverObject** In this robotic environment, the task of a quadrupedal robot [42] is to stand up and lean forward to reach a target position (purple markers need to reach green dots in Fig. 1b) without hitting an object visualized by the red cube representing the unsafe zone. The robot starts in a laying position as shown in the inset of Fig. 1b. As in the *Noisy-HalfCheetah* environment, Gaussian action noise is applied to mimic real-world perturbances.

### 4.1 Algorithmic Choices and Training Details

For model-predictive planning we use the CEM implementation from Pinneri et al. [31]. Further details about hyperparameters can be found in Suppl. A.2. For planning, we use the same architecture for the ensemble of probabilistic models, both in RAZER and in PETS. The only difference is that in RAZER we also forward propagate the mean state predictions in addition to the sampled state predictions. Further details can be found in Suppl. A.1.

### 4.2 Active Learning for Model Improvement

If model uncertainties are used for risk-averse planning, they are only meaningful if the model has the right training data. Only from good data can the parameters of the approximate noise model be learned correctly. In case of too little data, the agent might avoid parts of the state space due to an overestimation of the model uncertainties. On the other hand, the agent might enter unsafe regions

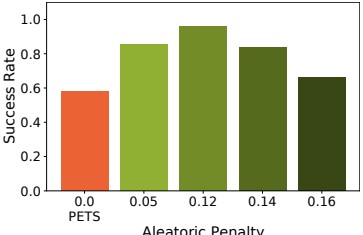

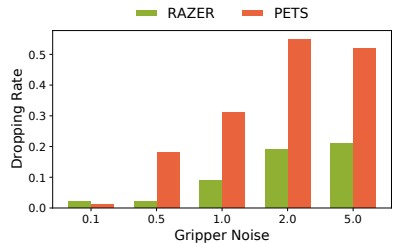

(a) *BridgeMaze* success depending on $w_{\mathfrak{A}}$ for 50 runs.

(b) Dropping rate in *Noisy-FetchPickAndPlace* for 100 runs.

Figure 4: Risk-averse planning in the face of aleatoric uncertainty yields higher success rates in noisy environments. For (b) we use ground truth models and a fixed aleatoric penalty weight $w_{\mathfrak{A}}$.

for which the uncertainties are underestimated. By adding the epistemic bonus to our domain-specific cost, the planner can actively seek states with high epistemic uncertainty, i.e. for which no or only little training data exists.

Figure 4a shows this active data gathering process for the *BridgeMaze* environment. PETS finds one particular solution to the problem of reaching the goal platform. It chooses the path over the safer, lower bridge rather than the dangerous middle path and the longer path via the upper bridge (Fig. 3b). Once, one solution is found, the model overfits to it without exploring any other parts of the state space. This is also reflected in the plateauing of the red curve in Fig. 3a.

In comparison, RAZER actively explores larger and larger parts of the state space with an increasing weight of the epistemic bonus (Fig. 3a). RAZER not only finds the easy solution found by PETS but also extensively explores other parts of the state space (Fig. 3c). To not get stuck at the middle bridge during exploration due to the inherent noise, it is important to separate between epistemic and aleatoric uncertainties. Only the former should be used for exploration. With enough data, our model can correctly capture the uncertainties of these states resulting in the epistemic uncertainty approaching zero.

## 4.3 Risk-Averse Planning

Once a good model is learned, it can be used for safe planning. What differentiates RAZER from PETS is that it makes explicit use of uncertainty estimates while in the latter uncertainties only enter planning by taking the mean over the particle costs and not differentiating between different sources of uncertainty.

**BridgeMaze.** Figure 4a shows the success rate of PETS and RAZER in the *BridgeMaze*. In both cases, we use the same model that was trained from data collected during a training run with $w_{\mathfrak{E}} = 0.05$. Hence, the model saw enough training data from all parts of the state space. The noise in the environment is tuned such that there is a chance to cross the bridge without falling. While in Fig. 3b PETS avoided this path because of an overestimation of the state's value due to a lack

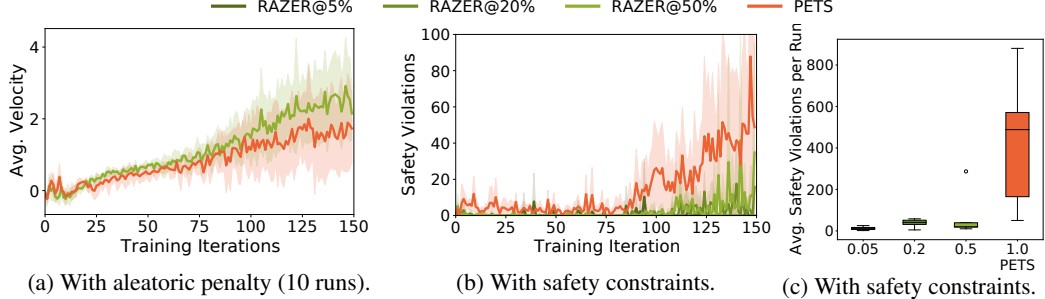

(a) With aleatoric penalty (10 runs).  (b) With safety constraints.  (c) With safety constraints.

Figure 5: *Noisy-HalfCheetah* environment (task lengths 300 steps) with learned models. At 150 iterations we have seen only 45k points. (a) Performance under noisy actions. By applying the aleatoric penalty, RAZER can navigate the uncertainties better – leading to higher returns faster. (b) Safety violations above a certain body height (simulating a low ceiling) for different values of $\delta$. In (c) the number of violations is averaged over the last 50 iterations (summed over 10 rollouts).

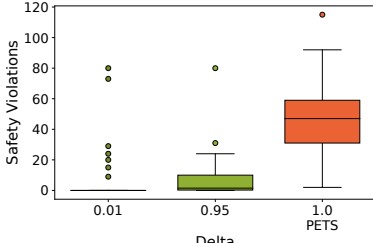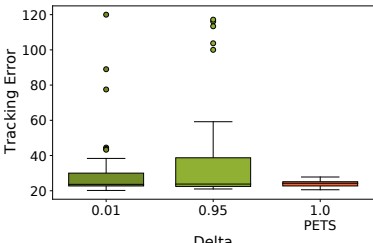

Figure 6: Safe planning vs. task-oriented planning in the *Solo8-LeanOverObject* environment with noisy actions. Left: number of safety violations for different values of $\delta$ (Eq. 11). Right: enforcing safety constraints causes slight reduction in tracking accuracy due to the fixed planning budget and the competing objectives of task and safety costs.

of training data and sometimes sees a chance to cross the bridge. However, these attempts are very likely to fail because of stronger winds that occur randomly, resulting in a success rate of only $58\%$. RAZER does not rely on sampling for the aleatoric part and can thus avoid risk. With a higher penalty constant the success rate increases up to $96\%$ but only as long as the agent is willing to take a risk at all. For large values of $w_{\mathfrak{A}}$ the agent becomes so conservative that it only moves slowly (decreasing reward in Fig. 4a).

**Noisy-HalfCheetah.** How does RAZER perform on the *Noisy-HalfCheetah* environment when models are learned from scratch? Without aleatoric penalty, the planner is optimistic. Risky situations are only detected if a failing particle is sampled. Thus, the noise is mostly neglected and the robot increases its velocity, gets destabilized, and ends up slower than with the aleatoric penalty (Fig. 5a).

**FetchPickAndPlace.** In this environment, a 7-DoF robot arm should bring the box to a target position – starting and target positions are at the opposite sides of the table. The shortest path is to lift the box and move in a straight line to the target. However, with noise applied to the gripper action, there is a certain probability to drop the box along the way. When penalizing aleatoric uncertainty, this is avoided and also fewer trajectory samples are "wasted" in high-entropic regions, as presented in Fig. 1a. Figure 4b shows the number of times the box is dropped on the table depending on the aleatoric penalty. RAZER adopts a cautious behavior, preferring to slide the box on the table and lifting it only in the area without action noise, maintaining a dropping rate lower than 20%, even when considerable noise is applied.

### 4.4 Planning with External Safety Constraints

**Noisy-HalfCheetah:.** We consider a safety constraint on the height of the body above ground simulating a narrow passage. Figure 5b shows the number of safety violations. Note that PETS has the same penalty cost for hard violations.

**Solo8-LeanOverObject:.** In this experiment, the robot has to move to two target points with its front and rear of the trunk while avoiding entering a specified rectangular area (fragile object). The front feet are fixed. To track the points, the robot has to lean forward, such that it can lose balance due to noisy actions. In contrast to PETS, RAZER successfully manages to satisfy the safety constraints almost always as shown in Fig. 6. However, satisfying the safety constraint comes with the cost of reduced tracking accuracy.

## 5 Conclusion

In this work, we have provided a methodology to separate uncertainties in stochastic ensemble models (PETSUS) which can be used as a tool to build risk-averse model-based planners that are also data-efficient and enforce safety through probabilistic safety constraints (RAZER). This type of risk-averseness can be achieved by a simple modification of the cost function in form of uncertainty penalties in zero-order trajectory optimizers.

Furthermore, the separation of uncertainties allows us to do proper exploration via epistemic bonus which benefits generalization of the model.As future work, it would be of interest to see this approach applied to a proper transfer learning setting from simulations to real systems, where risk-averseness combined with exploratory behavior is crucial for efficient learning and safe operation.

## Acknowledgments

This work was supported by the German Federal Ministry of Education and Research (BMBF): Tübingen AI Center, FKZ: 01IS18039B. The authors thank the International Max Planck Research School for Intelligent Systems (IMPRS-IS) for supporting Marin Vlastelica and Sebastian Blaes, and the Max Planck ETH Center for Learning Systems for supporting Cristina Pinneri.

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
