# OpenReview forum: "Risk-Averse Zero-Order Trajectory Optimization"
_robot-learning.org/CoRL/2021/Conference — CoRL2021 Poster_

### Official Review · Reviewer_Rk2o · 2021-07-23

**Originality:** Good
**Technical Quality:** Good
**Clarity Of Presentation:** Good
**Impact:** 3

**Recommendation:**

Weak Accept: I recommend accepting the paper, but will not argue for my recommendation if the majority of other reviewers have a different opinion.

**Summary:**

The paper is proposing a risk-aware CEM-based MPC method. The main machinery is extending the PETS model with direct consideration of the uncertainty terms (both epistemic and aleatoric) in the loss function. Although the proposed method is general in a sense it can be applied to an arbitrary zeroth-order optimizer, only CEM is considered in the experiments.

**Issues:**

- Ablation study for penalty weights

- Sensitivity analysis for penalty weights

- More consistent notation


**Reviewer Expertise:**

Very good: Comprehensive knowledge of the area

**Strengths And Weaknesses:**

The proposed method is technically sound. The consideration of uncertainties is critical for learning and authors do a good job in defining technically sensible formulations of uncertainty.

The application area of safety-critical MPC is a great match to the venue and also an important problem.

The Paper is well-written with some minor issues.

One major issue I see is the lack of ablation studies. It is not clear to me whether the separation of different uncertainties actually helps or not. Somewhat related, there are lots of penalty weights that need to be defined as hyper-parameters and they seem to impact the performance significantly. I believe this would warrant a discussion on how these should be chosen as well as an empirical analysis of the sensitivity to these penalty weights.

Minor but important: I think the notation is not entirely consistent. Frak and Cal fonts are used in a rather inconsistent manner. I don't know what frak font represents in this paper.

**Summary Of Recommendation:**

The paper is proposing an interesting algorithm and it is technically sound. I think it is a rather straightforward extension of PETS but it is an important and required extension. My only concern is the lack of ablation study and sensitivity analysis for penalty terms.

----

I thank the authors for their rebuttal. It answers my concerns. I believe the paper is definitely interesting and can be accepted. I will keep the weak accept score

---

> ### Author Response · Authors · 2021-08-23
> **Response to Reviewer Rk2o**
>
> Dear Reviewer Rk2o, we thank you for your detailed and thorough review.
>
> > One major issue I see is the lack of ablation studies. It is not clear to me whether the separation of different uncertainties actually helps or not. Somewhat related, there are lots of penalty weights that need to be defined as hyper-parameters and they seem to impact the performance significantly. I believe this would warrant a discussion on how these should be chosen as well as an empirical analysis of the sensitivity to these penalty weights.
>
> We provide a general answer to this question above.  Sensitivity studies are provided in figs 3a, 4, 5 and 6. For the safety constraints the parameter needs to be larger than  any “normal” cost ($c_{max}$) and involves no tuning.
>
> > Minor but important: I think the notation is not entirely consistent. Frak and Cal fonts are used in a rather inconsistent manner. I don't know what frak font represents in this paper.
>
> Thank you for pointing this out. We have also checked for further inconsistencies and removed them. We now use the frak font only to denote aleatoric, epistemic and safety - $\mathfrak{A}$, $\mathfrak{E}$ and $\mathfrak{S} respectively. The cal font is used as usual now.
>
> > Sensitivity analysis for penalty weights
>
> Sensitivity analysis for the different cost terms are provided in figures 3a, 4, 5 and 6. Can you provide more details of what is missing here?

---

### Official Review · Reviewer_pSNt · 2021-07-23

**Originality:** Good
**Technical Quality:** Good
**Clarity Of Presentation:** Fair
**Impact:** 3

**Recommendation:**

Weak Accept: I recommend accepting the paper, but will not argue for my recommendation if the majority of other reviewers have a different opinion.

**Summary:**

This paper proposes a method for separating aleatoric and epistemic uncertainties in learned dynamic models of noisy robotic systems. The paper also proposes a method for using these learned models in risk-averse model-based planning. The methods are demonstrated on a variety of simulation examples, and outperform the existing PETS method.

**Issues:**

I've included both positive notes that I took while reading as well as issues that I think could be improved.

Abstract
- short and clear, I like it

Introduction and Related Work
- Off the top of my head, I don't know what "zero-order" MPC is. To me, "first-order" means gradient descent for trajectory optimization, and "second-order" means also using a Hessian; so zero-order must mean a sampling-based method with no gradient information? Might be helpful to clarify this for less familiar readers.
- I'm very curious to see how this work performs uncertainty propagation.

Method
- I'm curious if this paper will make any statements about how much data is needed to fit $f$.
- The hypothesis that "accounting for uncertainty... is essential to develop risk-averse policies" feels tautological. Risk-aversion IS the process of accounting for uncertainty. That is, this statement strikes me as more like a definition than a hypothesis.
- The discussion of picking apart aleatoric and epistemic uncertainty is really interesting.
- The notation $\mathbb{X}$ has not been defined -- presumably this is supposed to be the whole state space.
- The simplification to enable computation seems drastic, since it's a subset of $\mathbb{R}^2$! The state can only be 2-D? That seems to contradict the Fetch arm and HalfCheetah examples in Fig. 1. I think the unsafe set needs to be described much more clearly; I guess this is supposed to mean that there can be many unsafe sets, and each one is a 2-D box? It would be clearer to just say the constraint set is (a union of) compact box/es in the state space.
- The fun acronyms for the proposed method (PETSUS,RAZER) should be stated in the introduction to prime the reader for these terms later on.

Experiments
- The first mention of PETS in this section should include a citation
- I guess the epistemic uncertainty cost weight in (10) is on some sort of schedule, starting out high and decreasing over time? If this isn't the case, it would be good to explain how and why RAZER transitions from exploration to exploitation.
- How long did training take? How many times was each method run in each environment? Is the proposed method able to run both training and testing in real time?
- The descriptions in Sec. 4.2 are a bit incoherent (some run-on sentences, unclear explanations of results).
- The figures are much more clear than the text, though!

Discussion
- It would be good to title this section as "Conclusion"
- This section is very clear and highlights the contribution of the proposed method!

**Reviewer Expertise:**

Good: General knowledge of the area

**Strengths And Weaknesses:**

STRENGTHS
- The main strength of this paper is Sec. 3.4: the separation of aleatoric and epistemic uncertainties from noisy data.
- The experimental results seem convincing that the method works.

WEAKNESSES
- I think the idea of a 1-step Gaussian nonlinear dynamics model is really odd; it seems to totally ignore the fact that we can and should model the parts of the dynamics that we actually know. I guess the method works, but it would be great to have a much clearer justification of why this Gaussian ensemble is a good model.
- The experiment section is written in a very confusing way. It's unclear that the experiments are thorough (e.g., when RAZER has a 96% success rate, I have no idea how many trials were conducted), though some information is related through the x-axis labels of the figures.

**Summary Of Recommendation:**

I think there is a lot of implementation information missing from this paper that would make it much easier to assess whether or not the method is practical and interesting; basically, the paper is pretty unclear about what it does. I think that, if the authors are able to clarify my concerns in the "weaknesses" section above, and the "issues" section below, then I'd be happy to accept this paper.

Final review/summary: The authors have addressed my concerns and the paper seems stronger now. I already felt "accept" was in order, so I think my current recommendation stands.

---

> ### Author Response · Authors · 2021-08-23
> **Response to Reviewer pSNt**
>
> Dear Reviewer pSNt, we thank you for your detailed and thorough review.
>
> > I think the idea of a 1-step Gaussian nonlinear dynamics model is really odd;
>
> We fully agree with the reviewer, that, if possible, analytical models should be used. Our choice for fully learned models is twofold. First and foremost, this work demonstrates the usefulness of separating epistemic and aleatoric uncertainties in learned models. In our opinion, adding complexity to the model only adds complexity to the paper without helping to support the main claims. Second, we are generally interested in complex, dynamical systems with contacts. For these types of systems, it is not trivial or often not even possible to write down analytical models. For the sake of general applicability of our method, we decided to concentrate on fully learned models.
>
> > The experiment section is written in a very confusing way. It's unclear that the experiments are thorough (e.g., when RAZER has a 96% success rate, I have no idea how many trials were conducted) ...
>
> We added a subsection to the experiment section explaining the architectural choices (4.1 (Algorithmic choices) as well as Suppl. A.1). We also added further details regarding the number of runs we average over.
>
> > Off the top of my head, I don't know what "zero-order" MPC is. To me, "first-order" means gradient descent for trajectory optimization, and "second-order" means also using a Hessian; so zero-order ..
>
> Yes, you are absolutely right. To make it more clear, we changed
> “This motivates the usage of zero-order methods, such as the Cross-entropy Method (CEM) [...]”
> to
> “This motivates the usage of zero-order, i.e gradient-free or sample-based, methods, such as the Cross-entropy Method (CEM) [...]” in the introduction.
>
> > I'm very curious to see how this work performs uncertainty propagation.
>
> We use a particle-based sampling approach to sample from the non-trivial distribution over trajectories defined by the model, similarly to the approach proposed in PETS. In comparison to PETS, where they consider only the expected cost of the particle-sampled trajectories (ie. taking the average), we also address the uncertainty that we estimate by moment matching of sampled particles. We have added a small clarification to section 3.2, but generally refer to the PETS paper for details.
>
> > I'm curious if this paper will make any statements about how much data is needed to fit $f$
>
> So far, we can only report empirical values from our experiments. Fpr bridge-maze we see that $10^5$ data points are sufficient (we are looking into reporting values with less data). The results in Fig 5 show the performance of half-cheetah with increasing data amount. We also added details of training to the appendix.
>
> > The hypothesis that "accounting for uncertainty... is essential to develop risk-averse policies" feels tautological. Risk-aversion IS the process of accounting for uncertainty. ...
>
> True, the statement seems tautological, but approaches based on sampling and taking the average (such as PETS) can also be risk-averse in cases where high-cost samples are able to shift the mean cost enough. This is why we felt that we need to make this explicit, we can change this to be a remark rather than a hypothesis.
>
> > The notation X has not been defined -- presumably this is supposed to be the whole state space.
>
> We added the definition of X in section 3.5.
>
> > The simplification to enable computation seems drastic, since it's a subset of R2! The state can only be 2-D? That seems to contradict the Fetch arm and HalfCheetah examples in Fig. 1. I think the unsafe set needs to be described much more clearly; I guess this is supposed to mean that there can be many unsafe sets, and each one is a 2-D box?...
>
> We agree with the reviewer. We rewrote this part to make it more clear.
>
> > The fun acronyms for the proposed method (PETSUS,RAZER) should be stated in the introduction to prime the reader for these terms later on.
>
> We agree. We introduced the acronyms in the introduction.
>
> > The first mention of PETS in this section should include a citation
>
> We added the citation.
>
> > I guess the epistemic uncertainty cost weight in (10) is on some sort of schedule, starting out high and decreasing over time? If this isn't the case, it would be good to explain how and why RAZER transitions from exploration to exploitation.
>
> There is no explicit scheduling imposed and the weight is fixed. However, effectively the epistemic bonus scales down automatically as we explore more parts of the state-space.
>
> > How long did training take? How many times was each method run in each environment? Is the proposed method able to run both training and testing in real time?
>
> We did not tune our method for speed specifically. However, due to the high parallelization we obtain real-time performance as reported in supplementary now. (see Suppl. A.3.)
>
> > The descriptions in Sec. 4.2 are a bit incoherent.
>
> We reworked that section to make it more clear.

---

> > ### Comment · Reviewer_pSNt · 2021-08-30
> > **Response to Authors**
> >
> > Thanks for considering all of my issues! I appreciate your thorough response.

---

### Official Review · Reviewer_RZzg · 2021-08-02

**Originality:** Good
**Technical Quality:** Good
**Clarity Of Presentation:** Very Good
**Impact:** 4

**Recommendation:**

Weak Accept: I recommend accepting the paper, but will not argue for my recommendation if the majority of other reviewers have a different opinion.

**Summary:**

This paper introduces a methodology that improves exploration for uncertainty-aware dynamics models using probabilistic ensembles of neural networks in CEM-MPC planning (zero-order trajectory optimization approach) by providing an explicit separation between aleatoric and epistemic uncertainty of the learned dynamics model. The approach follows closely the methodology of previous work, PETS [6], yet improves upon it by  (i) adding an autoregressive component in the dynamics models for forward prediction, (ii) introducing an entropy-driven approach to measure epistemic and aleatoric uncertainty in the ensemble of neural nets, (iii) uses the estimated aleatoric uncertainty to penalize an MPC cost function while adding epistemic bonuses estimated from the epistemic uncertainty. Further, due to the chosen zero-order optimization technique, safety constraints can also be considered by formulating them as probabilities of safety constraints violations and adding them as penalty to the cost function. The resulting algorithm, RAZER, is compared to PETS in 4 continuous state-action space experiments in simulated environments of increasing complexity. Results clearly show that the proposed methodological improvements (RAZER) outperform PETS in terms of exploration/state-space coverage, uncovering safe solutions in regions with little training data guided by the epistemic bonus and safety constraints.

**Issues:**

See Weaknesses.


Minor Errors:

-Page 5, Line 194 “we define the an aleatoric”

-Page 6, line 235 “risk-avers planning”

-Page 7, line 243 “the somewhat save lower”->safe

-Page 7, Line 271 “from scratch”?


**Reviewer Expertise:**

Good: General knowledge of the area

**Strengths And Weaknesses:**

Strengths:

-Principled methodology to uncover both epistemic and aleatoric uncertainties from learning dynamics models. While this is currently applied to zero-order trajectory optimization problems, such explicit separation can be used in other tasks involving predictions of uncertain dynamics systems.

-Consideration of safety constraints as a form of uncertainty in the cost function is a clever way of dealing with constraint satisfaction without explicitly having to guarantee constraints in the MPC optimizer.

-While the modifications to PETS might seem incremental, I believe the results are compelling as they show clearly improved exploration and state-space coverage over baseline.

Weaknesses:

-PETS uses the learned models within an CEM-MPC planner too. Here, it is claimed that an efficient implementation of an CEM-MPC planner from [31] was used. What is the main advantage/difference? Where is this efficiency reflected in the evaluation of results? Is [31] just a faster implementation of CEM-MPC? If so, how fast? Is PETS being implemented with this CEM-MPC planner too or with the original version? Analysis of such computational complexity is missing.

-While the inclusion of safety constraints as penalties in the cost is a nice trick to include safety in planning the type of constraints allowable, basically bounds on the states [a,b] where a<b might be too restrictive. Is there a way to include other types of constraints in the approach, such as polytopic constraints, which are often found in robotics problems involving contacts and safety?

-Including the constraints as cost in the CEM-MPC is not the only way to guarantee the satisfaction of safety constraints. There are other approaches that have done this, see relevant paper “Constrained Cross-Entropy Method for Safe Reinforcement Learning” from Wen and Topcu. It could be beneficial to discuss the differences with newer constrained-based approaches and how the current methodology could be extended to adopt them.

-Could you provide an intuition or rule-of-thumb as to how to define and leverage the weights in the cost function for the different penalties/bonuses? For example, what’s the resulting behavior if both aleatoric penalty and epistemic bonus are high and safety constraint is low, or vice-versa or any combination of the three. In the current submission, we are only given analyzes of either the change in weights of the aleatoric penalty or epimestic bonus or increase in safety constraint probability threshold, independently.

-As mentioned in the text, satisfying constraints reduces tracking accuracy, how can this be alleviated?


**Summary Of Recommendation:**

This paper presents interesting insights into the modeling of different types of uncertainties with neural networks and their use for trajectory-optimization. The work builds upon a previously introduced framework, yet the proposed improvements seem to provide compelling results, specifically in the improvements of state-space exploration while remaining within safe bounds. Further analysis of specific claims and decision choices can be beneficial (see weaknesses).

---

> ### Author Response · Authors · 2021-08-23
> **Response to Reviewer RZzg**
>
> Dear Reviewer RZzg, we thank you for your detailed and thorough review.
>
> > PETS uses the learned models within an CEM-MPC planner too. Here, it is claimed that an efficient implementation of an CEM-MPC planner from [31] was used. What is the main advantage/difference? Where is this efficiency reflected in the evaluation of results? Is [31] just a faster implementation of CEM-MPC? If so, how fast? Is PETS being implemented with this CEM-MPC planner too or with the original version? Analysis of such computational complexity is missing.
>
> First of all, we want to clarify that we used the same CEM implementation, i.e. the one from [31], for PETS and RAZER. The goal of this work is to study the benefits of disentangling the different types of uncertainties and how to use them during planning. Hence, we kept all other architectural choices the same between RAZER and PETS. We added a short subsection to section 4 to make this point more clear.
> We chose the CEM implementation from [31] because it promises a significant computation speed-up compared to the vanilla CEM implementation.
> Analysis of the computation complexity of the CEM implementation is out of the scope of our paper and we refer the reviewer to [31] for more details regarding the computational complexity.
>
> > While the inclusion of safety constraints as penalties in the cost is a nice trick to include safety in planning the type of constraints allowable, basically bounds on the states [a,b] where a<b might be too restrictive. Is there a way to include other types of constraints in the approach, such as polytopic constraints, which are often found in robotics problems involving contacts and safety?
>
> More general constraints are a good point. We present the general form for arbitrary constraints in the paper already and mention Monte-Carlo sampling for computing the probability of a constraint violation. However, this might be too slow. We chose the isotropic Gaussian with box constraints case as proof of concept for computational simplicity since the focus of this work is not tackling various probabilistic safety constraint regions.
> An interesting further investigation would be to see under which conditions/constraint types a good closed-form computation or fast approximation for the intersection/integral of the Gaussian is possible.
>
> > Including the constraints as cost in the CEM-MPC is not the only way to guarantee the satisfaction of safety constraints. There are other approaches that have done this, see relevant paper “Constrained Cross-Entropy Method for Safe Reinforcement Learning” ...
>
> Thank you very much for the pointer to this paper. We refer to it in the paper. However, we believe, their method can be similarly implemented by adding a cost proportional to a constraint violation (that is larger than all “normal” costs ($c_{max}$ see below Eq. 11). Thus, our approach is similar, but differs in that 1) we address probabilistic constraints 2) we deal with stochastic transition functions 3) we utilize a stochastic learned model of the dynamics.
>
> > Could you provide an intuition or rule-of-thumb as to how to define and leverage the weights in the cost function for the different penalties/bonuses? For example, what’s the resulting behavior if both aleatoric penalty and epistemic bonus are high and safety constraint is low, or vice-versa or any combination of the three. In the current submission, we are only given analyzes of either the change in weights of the aleatoric penalty or epimestic bonus or increase in safety constraint probability threshold, independently.
>
> We chose to independently do the sensitivity analysis because the individual cost terms serve distinct purposes.
>
> The safety term overtakes the cost once the probability threshold is exceeded, so the weight on this term could be in principle infinite, because we don’t want to compromise safety. We will add a more thorough discussion about the safety term weight in the manuscript.
>
> Regarding the aleatoric and epistemic terms, please see our general answer.
>
> > As mentioned in the text, satisfying constraints reduces tracking accuracy, how can this be alleviated?
>
> In fact, it is not too surprising that tracking accuracy drops for RAZER because of how the experiment is constructed. The tracking targets are placed in the environment such that the robot has to come very close to the safety violation region (see fig. 1b). Because RAZER is aware of its own stochastic nature, it has to keep a certain margin to the safety violation region in order to avoid safety violations with a pre-specified probability, resulting in a higher tracking error. This effect can be mitigated with a higher sample size. PETS on the other hand is only concerned with minimizing the tracking error and therefore can use all computational resources to optimize this objective. We will make this more clear in the experiments section.

---

> > ### Comment · Reviewer_RZzg · 2021-09-04
> > **Response to Rebuttal**
> >
> > I want to thank the authors for addressing my comments in the rebuttal and including more details of the algorithmic choices in the revised manuscript. I appreciate their clarifications and work done to improving the submission. I would be happy to recommend accepting this paper if the other reviewers agree.

---

### Author Response · Authors · 2021-08-23
**General Answer**

We thank the area chair for the extended meta-review and all of the reviewers for constructive feedback. For answers specific to each reviewer, see the respective thread.

> The reviewers highlighted the fact that hyperparameters in the approach may not be entirely straightforward to choose and that the paper should provide rules of thumb on how to choose them. In addition, further details on the experimental evaluation would be helpful (see Reviewer pSNT and RZzg’s comments).

We added subsection 4.1 (Algorithmic choices) as well as Suppl. A.1 to the paper, providing more details regarding the architecture and hyperparameter choices. We added the number of runs over which means and standard deviations are computed, making the evaluation process more transparent.

> One concern with the experimental evaluation is the lack of thorough ablation studies. Given that the primary hypothesis of the paper is that epistemic and aleatoric uncertainty should be separately dealt with, the experimental evaluation should thoroughly demonstrate this via ablation studies.

Regarding ablation studies, so far we provide ablations for all of the introduced terms in the cost (epistemic bonus, aleatoric penalty and safety penalty) by comparing to the baseline (MPC-CEM) algorithm that uses the same model as we do, but with different utilization of the samples. We have also provided sensitivity analysis for all of the terms introduced, aleatoric penalty (Fig. 4.a, epistemic bonus (Fig 3.a), safety delta threshold (Fig 4.b, 4.c, Fig. 6).

Aleatoric and epistemic terms serve fundamentally different purposes. In a safe environment we want to completely focus on the epistemic term, whereas in evaluation (on a real robot for example), we want to be conservative and use the aleatoric uncertainty as penalty. We plan to analyze the interplay between these terms in a sim2real setting in future work, where it’s essential to balance them at evaluation time where we want to gather data to improve the model. We will make this more explicit.

In response to the comments, we have uploaded a revision that takes your suggestions into account with clarifications and improvement of consistency.

If you require more clarification, please don't hesitate to let us know.

---

### Meta-Review · Area_Chair_naAx · 2021-08-02

**Recommendation:** Accept (Poster)
**Confidence:** 4

**Metareview:**

This paper proposes an approach for incorporating risk in zero-order (i.e., derivative-free) trajectory optimization. In particular, the method uses an ensemble of stochastic neural networks in order to estimate aleatoric and epistemic uncertainty in the robot dynamics model. The approach seeks to be optimistic in the face of epistemic uncertainty and pessimistic in the face of aleatoric uncertainty. The resulting model predictive control scheme is demonstrated on different robot control examples.

Strengths:
+ The reviewers agree that the key idea of separating aleatoric uncertainty and epistemic uncertainty and dealing with these in different ways is a promising one.
+ The reviewers generally agree that the experimental evaluation in the paper is thorough and convincing (with one point of concern noted below).
+ The reviewers agree that the paper is well written and clear overall.

Weaknesses:
- The reviewers highlighted the fact that hyperparameters in the approach may not be entirely straightforward to choose and that the paper should provide rules of thumb on how to choose them. In addition, further details on the experimental evaluation would be helpful (see Reviewer pSNT and RZzg’s comments).
- One concern with the experimental evaluation is the lack of thorough ablation studies. Given that the primary hypothesis of the paper is that epistemic and aleatoric uncertainty should be separately dealt with, the experimental evaluation should thoroughly demonstrate this via ablation studies.

Suggestions:
I urge the authors to consider the reviewers’ feedback  and the primary points highlighted above. The main suggestions are to: (i) add further details on the approach, hyperparameters, and experimental evaluation (in the places suggested by the reviewers), and (ii) address the point regarding ablations.

----- Post rebuttal -----

In the rebuttal period, the authors have addressed the primary concerns reviewers had raised by (i) adding more details on algorithmic choices, and (ii) clarifying the role of taking into account epistemic and aleatoric uncertainty. The reviewers are now in agreement that the paper makes a good contribution and should be accepted.

---

### Decision · Program_Chairs · 2021-09-13

**Decision:**

Accept (Poster)

**Comment:**

This paper proposes an approach for incorporating risk in zero-order (i.e., derivative-free) trajectory optimization. In particular, the method uses an ensemble of stochastic neural networks in order to estimate aleatoric and epistemic uncertainty in the robot dynamics model. The approach seeks to be optimistic in the face of epistemic uncertainty and pessimistic in the face of aleatoric uncertainty. The resulting model predictive control scheme is demonstrated on different robot control examples.

Strengths:
+ The reviewers agree that the key idea of separating aleatoric uncertainty and epistemic uncertainty and dealing with these in different ways is a promising one.
+ The reviewers generally agree that the experimental evaluation in the paper is thorough and convincing (with one point of concern noted below).
+ The reviewers agree that the paper is well written and clear overall.

Weaknesses:
- The reviewers highlighted the fact that hyperparameters in the approach may not be entirely straightforward to choose and that the paper should provide rules of thumb on how to choose them. In addition, further details on the experimental evaluation would be helpful (see Reviewer pSNT and RZzg’s comments).
- One concern with the experimental evaluation is the lack of thorough ablation studies. Given that the primary hypothesis of the paper is that epistemic and aleatoric uncertainty should be separately dealt with, the experimental evaluation should thoroughly demonstrate this via ablation studies.

Suggestions:
I urge the authors to consider the reviewers’ feedback  and the primary points highlighted above. The main suggestions are to: (i) add further details on the approach, hyperparameters, and experimental evaluation (in the places suggested by the reviewers), and (ii) address the point regarding ablations.

----- Post rebuttal -----

In the rebuttal period, the authors have addressed the primary concerns reviewers had raised by (i) adding more details on algorithmic choices, and (ii) clarifying the role of taking into account epistemic and aleatoric uncertainty. The reviewers are now in agreement that the paper makes a good contribution and should be accepted.